# DNA Fragmentation in Viable and Non-Viable Spermatozoa Discriminates Fertile and Subfertile Subjects with Similar Accuracy

**DOI:** 10.3390/jcm9051341

**Published:** 2020-05-04

**Authors:** Monica Muratori, Giulia Pellegrino, Giusi Mangone, Chiara Azzari, Francesco Lotti, Nicoletta Tarozzi, Luca Boni, Andrea Borini, Mario Maggi, Elisabetta Baldi

**Affiliations:** 1Department of Experimental and Clinical Biomedical Sciences “Mario Serio”, Center of Excellence DeNothe, University of Florence, 50139 Florence, Italy; giulia.pellegrino3004@gmail.com (G.P.); francesco.lotti@unifi.it (F.L.); mario.maggi@unifi.it (M.M.); 2Pediatric Section, Department of Health Sciences, University of Florence and Anna Meyer Children’s University Hospital, 50139 Florence, Italy; giusi.mangone@meyer.it (G.M.); chiara.azzari@meyer.it (C.A.); 39.baby, Family and Fertility Center, 40125 Bologna, Italy; nicoletta.tarozzi@9puntobaby.it (N.T.); andrea.borini@9puntobaby.it (A.B.); 4Clinical Trials Center, AOU Careggi, 50139 Florence, Italy; bonil@aou-careggi.toscana.it; 5Department of Experimental and Clinical Medicine, Center of Excellence DeNothe, University of Florence, 50139 Florence, Italy; elisabetta.baldi@unifi.it

**Keywords:** sperm DNA fragmentation, viable spermatozoa, male infertility

## Abstract

Sperm DNA fragmentation (sDF) negatively affects reproduction and is traditionally detected in total sperm population including viable and non-viable spermatozoa. Here, we aimed at exploring the ability of DNA fragmentation to discriminate fertile and subfertile men when detected in viable (viable sDF), non-viable (non-viable sDF), and total spermatozoa (total sDF). We revealed sDF in 91 male partners of infertile couples and 71 fertile men (max 1 year from natural conception) with LiveTUNEL coupled to flow cytometry, able to reveal simultaneously DNA fragmentation and cell viability. We found that the three sDF parameters discriminated fertile and subfertile men with similar accuracy and independently from age and basal semen parameters: AUCs (area under the curves) (95% CI) were: 0.696 (0.615–0.776), *p* < 0.001 for total sDF; 0.718 (0.640–0.797), *p* < 0.001 for viable sDF; 0.760 (0.685–0.835), *p* < 0.001 for non-viable sDF. We also found that total and non-viable but not viable sDF significantly correlated to age and semen quality. In conclusion, the three sDF parameters similarly discriminated fertile and subfertile men. Viable spermatozoa with DNA fragmentation are likely cells able to fertilize the oocyte but failing to properly support subsequent embryo development. Non-viable sDF could be a sign of a subtler damage extended beyond the non-viable cells.

## 1. Introduction

Male infertility affects 7% of men and in up to 40% of cases, the causes of such a condition is not fully understood. In addition, in Western societies a progressive deterioration of spermatogenesis occurred in the last four decades [1] along with an increased incidence of other pathologies of the male reproductive system [2]. It is believed that changes in lifestyle and the increasing exposure to environmental pollution have a great role in the deterioration of semen quality and in the increase of male infertility occurred in the last forty years [3,4,5]. Among the alterations provoked by noxious lifestyle factors and by pollution on male fertility, sperm DNA fragmentation (sDF) plays an important role [6]. Sperm DNA fragmentation originates from testicular processes such as apoptosis or impairment of chromatin maturation of spermatids as well as from oxidative attack occurring in the genital tract [7]. It is well known that sDF negatively affects reproduction, by delaying time to natural pregnancy, by decreasing the probability to achieve pregnancy [8,9,10], and by increasing the risk of miscarriage [11,12,13]. However, the literature on these topics is controversial and we are far from understanding how best to introduce the assessment of sDF into the male infertility work-up [14].

It has been demonstrated that a large part of ejaculated spermatozoa with DNA fragmentation are non-viable [15]. Importantly, our group showed that the origin of sDF could be different when considering viable and non-viable spermatozoa [7]. In particular, only in viable spermatozoa was DNA fragmentation significantly associated to signs of oxidative damage to the membrane and to DNA, whereas no association was found in the bulk of semen sample [7], containing mainly non-viable DNA fragmented spermatozoa [15]. These results suggest that oxidative stress plays a main role in generating sDF in the viable sperm fraction, where the onset of DNA breakage occurs, likely, during the transit in the male genital tracts shortly before ejaculation [7] and when most of the antioxidant defenses are lost [16]. Oxidative stress appears also to be the link between the increased levels of sDF and several noxious conditions (including obesity, a smoking habit, environmental exposure to pollutants and to radiation) [17,18], diseases (such as diabetes [19] and varicocele [20]) and aging [21], all reported to increase the levels of semen reactive oxygen species. For all these reasons, evaluation of sDF in the viable sperm fraction may give clues in understanding the link between DNA damage, oxidative attack, and male subfertility. In addition, we should consider that non-viable spermatozoa have no chance to reach the oocyte and thus, albeit DNA damaged, to affect reproduction outcomes. Conversely, DNA damaged viable spermatozoa can fertilize the oocyte and alter embryo development if the damage is not repaired by the oocyte [22,23,24]. According to this speculation, DNA breaks in the viable spermatozoa (viable sDF) should better discriminate fertile and infertile men than sDF revealed in the bulk of spermatozoa (total sDF). In a previous study, we demonstrated that sDF in a sperm subpopulation (named brighter population) containing viable DNA fragmented spermatozoa was more predictive of the male fertility status than sDF detected in the whole sperm population [10]. In the present study, we challenge the hypothesis that viable sDF is more predictive than the total one which is presently evaluated by the available techniques but shows limitations as screening test for natural pregnancy [14]. To this aim, we used a flow cytometric technique able to reveal simultaneously sperm viability and DNA fragmentation (LiveTUNEL) and thus to determinate sDF in both viable and non-viable spermatozoa (non-viable sDF). Viable, non-viable, and total sDF were evaluated and compared in 71 fertile subjects and 91 male partners of infertile couples.

## 2. Material and Methods

### 2.1. Subjects and Sample Collection and Preparation

Fertile subjects (*n* = 71) were men who had recently fathered a child (max 1 year from natural conception), whereas patients (*n* = 91) were subfertile subjects recruited consecutively from male partners of infertile couples who presented at 9.baby center (Bologna, Italy) for routine semen analysis. Female factors of infertility in these couples were unknown. We excluded subjects with recent pharmacological therapies or high fever and with a sperm number < 2 × 10^6^ millions/ejaculate and leukocytospermia > 1 million/mL. Fertile men who obtained pregnancy by Assisted Reproduction Techniques were also excluded. Semen samples from subfertile and fertile men were collected during the period 2017–2019. In both groups of subjects, semen samples were collected by masturbation after 2–7 days abstinence period and analyzed for sperm number, concentration, motility, and morphology according to WHO procedures [25]. Briefly, spermatozoa were appropriately diluted with fixative (0.3% formalin) and counted by improved Neubauer hemocytometer; motility was evaluated by distinguishing and calculating the percentage of progressive, non-progressive and immotile spermatozoa; morphology was evaluated after Diff-Quick staining. Motility and morphology were evaluated scoring 200 spermatozoa/sample. From here on, sperm number, concentration, motility, and morphology evaluated according to WHO procedures are referred to as conventional semen parameters. The study was conducted in accordance with the Declaration of Helsinki and was approved by the internal institutional review board (11 October 2016) of 9.baby center (Bologna, Italy). Informed consent was obtained by recruited subjects.

### 2.2. LiveTUNEL

For detection of DNA breaks both in total and viable spermatozoa, we performed a double staining with TUNEL (terminal deoxynucleotidyl transferase (TdT)-mediated fluorescein-dUTP nick end labeling) and LIVE⁄ DEAD Fixable Far Red Dead Cell Stain Kit (L10120, Life Technologies, Paisley, UK) (from herein indicated as LiveTUNEL, [26]). The L10120 binds dead cells stably and the binding is kept after sample fixation and permeabilization. Labelling with L10120 (diluted 1:10,000) was performed in fresh semen samples (2–10 × 10^6^ spermatozoa) after washing twice with HTF medium and incubating for 1 h at RT, in the dark, in 500 μL of phosphate buffered saline (PBS). After further washing, the samples were fixed (500 μL of 4% paraformaldehyde in PBS, pH 7.4, for 30 min at RT), and labelled by TUNEL as described elsewhere [27]. Briefly, after twice washing with PBS/bovine serum albumin 1%, samples were permeabilized (0.1% Triton X-100 in 100 µL of 0.1% sodium citrate for 4 min in ice), washed again, and incubated (1 h at 37 °C in the dark) in 50 µL of labelling solution (supplied with the in situ Cell Death Detection Kit, fluorescein, Sigma–Aldrich, Milan, Italy) containing the TdT enzyme. Then samples were washed twice, suspended in PBS and stained with DAPI (1 μg/mL, 15 min in the dark at RT) until acquisition with flow cytometry. For each test sample, a negative control was also prepared by omitting TdT.

### 2.3. Flow Cytometric Analyses

Flow cytometric analyses were conducted by a FACSAria II flow cytometer (BD Biosciences, Franklin Lakes, NJ, USA) equipped with a violet laser at 405 nm, a blue laser at 488 nm and a red laser at 633 nm for excitation. Samples were filtered by 50 μm Syringe Filcons (BD Biosciences, Franklin Lakes, NJ, USA) immediately before acquisition. To detect blue (DAPI), green (FITC), and far red (L10120) fluorescence, we used photomultiplier tubes equipped with, respectively, 450/40, 530/30, and 660/20 BP filters.

The BD FACSdiva Software (BD Biosciences, Franklin Lakes, NJ, USA) was used for acquisition and data analysis. For flow cytometric data acquisition and analysis, we drew the characteristic Forward Scatter/Side Scatter flame-shaped region containing spermatozoa and semen apoptotic bodies [28,29] (P1 region in upper panel of Figure 1A). Then, within P1 region, we gated the DAPI-labelled events (i.e., spermatozoa [7], P2 region in the middle panel of Figure 1A). Finally, a further region was drawn within the P2 region delimitating L10120 negative events (i.e., viable spermatozoa, P3 region in lower panel of Figure 1A). For each sample, 8000 viable sperm (in P3 region) were recorded. For data analysis, quadrants were set in the L10120/TUNEL dot plot of the negative controls, and then copied in the corresponding test samples. However, we noticed that, in several semen samples, the threshold for TUNEL set in non-viable spermatozoa was not suitable for the viable ones (and vice versa), as the non-specific TUNEL fluorescence in the former can be higher than that in the latter (left panel, Figure 1B). In the 162 examined subjects, a threshold including a median value (interquartile range) of 99.0% (99.8–99.2) of the viable spermatozoa, includes only 94.75% (88.7–97.1) of the non-viable spermatozoa. Hence, in the L10120/TUNEL dot plot of the negative controls, TUNEL threshold (solid line, Figure 1B) was set independently for viable and non-viable spermatozoa, thus including about 99% of the events both of the non-viable (quadrant Q1 in left panel of Figure 1B) and of the viable spermatozoa (quadrant Q3 in left panel of Figure 1B). Viable sDF was calculated as a percentage of the viable spermatozoa (Q4/Q3+Q4, right panel, Figure 1B). Similarly, non-viable sDF was calculated as a percentage of non-viable (Q2/Q2+Q1, right panel, Figure 1B) spermatozoa. Total sDF was calculated simulating the traditional TUNEL assay which does not discriminate between viable and non-viable spermatozoa. Hence, after gating all DAPI positive population (P2 region, middle panel, Figure 1A), i.e., total sperm population, we set a threshold in the negative control, including about 99% of the spermatozoa and then copied it in the corresponding test samples (Figure 1C).

### 2.4. Statistical Analysis

Data were analyzed by Statistical Package for the Social Sciences (SPSS 25) for Windows (SPSS, Inc Chicago, IL, USA) and the SAS System 9.2. Shapiro–Wilk test was used to test all variables for normal distribution. As all the variables, except for one, showed a non-normal distribution, the results were expressed as median (interquartile range). Comparison of sDF, age, abstinence time, semen volume, and conventional semen parameters between fertile men and patients were assessed by the Mann–Whitney U-test. Bivariate correlations among the three fractions of sDF and semen parameters, abstinence time and age were evaluated by calculating the Spearman’s correlation coefficient (*r*). The ability of sDF to identify fertile men and patients, was assessed by the AUC (area under curve) of ROC (Receiver Operating Characteristic) curves, calculated by means of the univariate and multivariate logistic regression model using the different parameters of sDF as “test variables”, and “fertile versus patient” as “state variable”. For multivariate analyses, conventional semen parameters, abstinence time and age were introduced as binary covariates into the model, besides the single tested sDF parameters.

## 3. Results

### 3.1. LiveTUNEL

To compare sDF in patients and fertile men we used LiveTUNEL, a flow cytometric technique, able to detect DNA fragmentation in viable and non-viable spermatozoa, other than in total (i.e., viable + non-viable) ones [26] (Figure 1B). When comparing traditional TUNEL with LiveTUNEL, it is important to consider the fact that traditional TUNEL can underestimate sDF due to the fact that non-specific TUNEL fluorescence can be higher in non-viable than in viable spermatozoa (Figure 1B, upper panels), increasing the threshold over which cells are considered DNA fragmented (dotted line in Figure 1B) and thus masking a fraction of viable sDF (in grey in Figure 1B). Since we were interested in comparing the new sDF parameters (viable and non-viable sDF) to sDF as determined by traditional TUNEL, total sDF was calculated by gating total spermatozoa without discriminating viable and non-viable spermatozoa (Figure 1C).

### 3.2. sDF in Fertile Men and in Patients

Table 1 reports the values of conventional semen parameters, semen volume, abstinence time, and age in the two groups of subjects, showing that patients had a reduced sperm morphology and motility (progressive and total) and a slightly higher abstinence time with respect to fertile men. In addition, patients showed higher values of total, viable, and non-viable sDF than fertile men (Figure 2).

We also studied the relationship between sDF parameters and age, conventional semen parameters and abstinence time of the recruited subjects. Total sDF resulted positively correlated with age and abstinence time and negatively correlated with sperm morphology and progressive and total motility (Table 2). Similar correlations with age, abstinence time, sperm morphology, and motility were found also considering non-viable sDF (Table 2, Figure 3). Interestingly, viable sDF did not correlate either with age or abstinence time or with any sperm parameter (Table 2 and Figure 3). Hence, it was independent from semen quality. A strict correlation was found between viable and non-viable sDF (*r* = 0.698, *p* < 0.001, *n* = 162). In addition, as expected, a strict correlation was found also between total sDF and viable (*r* = 0.758, *p* < 0.001, *n* = 162) and non-viable sDF (*r* = 0.778, *p* < 0.001, *n* = 162).

### 3.3. SDF as a Predictor of Fertile versus Subfertile Status

The ability of sDF to discriminate patients from fertile men was analyzed by ROC curves, calculating the corresponding AUCs (Table 3). As shown, all the sDF parameters predicted fertile versus subfertile status, with the lower accuracy observed for total sDF.

We next evaluated whether the predictive ability of the different fractions of sDF were affected by semen quality, abstinence time, and age of recruited subjects. Using a multivariate analysis, we found that both viable sDF and non-viable sDF and total sDF, predicted fertile versus subfertile status independently from number, motility and morphology of spermatozoa, abstinence time, and age of the subjects (Appendix A).

## 4. Discussion

This study explored whether detecting sDF in viable spermatozoa can provide a better ability to predict fertile versus subfertile status, with respect to the traditional sDF parameter detected in the total sperm population. Indeed, we hypothesized that the DNA fragmented spermatozoa impacting natural reproduction are the viable ones. In fact, they should be able to reach the oocyte and to fertilize it [22], but, likely, they could fail in successfully supporting the subsequent embryo development [23,24]. Conversely, non-viable spermatozoa, albeit DNA fragmented, have no chance to reach the oocyte and, thus, to impact natural conception. Contrary to the expected, our study showed that the predictive ability of viable sDF to discriminate fertile men from patients was not better than that of total or even of non-viable sDF. Overall, this result suggests that elevated sDF levels are a general sign of some sperm alterations (at present unknown) impairing male fertility status (see below for further discussion about this point).

An important aspect of our study concerns the use of LiveTUNEL, a technique developed in our laboratory that allows simultaneous detection of DNA breakage in viable, non-viable, and total sperm populations. With this technique, we found values of viable sDF (mean±SD = 25.39 ± 16.75 in 91 patients) similar to those previously reported by our group [26] but higher than those reported with other techniques evaluating sDF in viable spermatozoa [15,30]. The reason is that our technique, at variance with previous ones, exclude both semen apoptotic bodies [29] and non-viable spermatozoa from the calculation of viable sDF [26], as inclusion of these parameters underestimates the value [28,31]. In addition, we show here that the non-specific fluorescence of non-viable spermatozoa may mask a fraction of viable DNA fragmented cells (Figure 1B). Thus, the threshold for detecting TUNEL positive spermatozoa should be established separately in viable and in non-viable cells as shown in Figure 1B.

As mentioned, the results of our study are quite surprising, as we found a similar ability of sDF in viable and non-viable spermatozoa to discriminate between fertile subjects and patients. The ability of viable sDF to predict fertile versus subfertile status can be easily explained considering that viable DNA fragmented spermatozoa can reach and fertilize the oocyte [22]. These spermatozoa, however, are likely failing in those embryo phases where the male genome starts to express [22,23,24]. A previous study from our group suggested that DNA breaks in viable spermatozoa may be acquired after spermiation during transit throughout the male genital tracts [7], where male gametes may encounter an oxidative environment [32]. The DNA damage in viable spermatozoa can arise also after ejaculation, after contact with seminal plasma [33,34] and during in vitro sperm processing [26,35]. The lack of correlation between viable sDF and conventional semen parameters or age of the subjects, showed in this study, appears in agreement with the post-testicular origin of viable sDF [7,33,34,35]. Semen quality mirrors mainly spermatogenesis and age strictly affects it [36], hence the lack of correlation with semen quality and age suggests that viable sDF mainly occurs after spermiation.

The ability of the amount of non-viable DNA-fragmented spermatozoa to distinguish fertile and subfertile men is difficult to explain. One possible explanation is that such amount reflects a broader impairment of spermatogenesis. This explanation is also reinforced by the fact that non-viable sDF correlates negatively with conventional semen parameters and positively with age (Figure 3, Table 2). This concept somehow reminds the “tip of iceberg” theory proposed by Evenson et al. [37] to explain why subjects with sDF over a threshold of 30% resulted infertile despite the remaining 70% of spermatozoa, apparently without DNA damage. Accordingly, the amount of non-viable sDF would represent the visible part of a subtler and more extended damage in the sperm population, responsible for in vivo infertility. This speculation is consistent with the strict association between viable and non-viable sDF reported in the present study. Such association could suggest that DNA damage in viable spermatozoa, albeit likely occurring following spermiation [7,33,34], is promoted mainly in spermatozoa with a vulnerability acquired during spermatogenesis. For instance, it is believed that impairment in chromatin maturation makes spermatozoa more susceptible to undergo DNA damage under a condition of oxidative stress [38].

Our study demonstrates that a fraction of non-viable spermatozoa does not exhibit DNA fragmentation (Figure 1B and Figure 2). This finding provides further evidence indicating that nuclei of non-viable spermatozoa are highly heterogeneous [28,39]. In previous studies [28,40], we reported that sperm nuclei may belong to two different sperm populations, showing a brighter and a dimmer intensity after nuclear staining. Brighter and dimmer spermatozoa are characterized by a different degree of chromatin condensation [39], likely explaining why dimmer sperm nuclei are less colored than the brighter ones [28]. Non-viable spermatozoa without DNA fragmentation reported here (Figure 1B and Figure 2) belong necessarily to the brighter population, as dimmer spermatozoa are all DNA fragmented [28]. In summary, non-viable spermatozoa appear to be at least composed by three subpopulations: (i) cells with DNA fragmentation and higher chromatin condensation (dimmer spermatozoa [28,39]); (ii) cells with DNA fragmentation and lower condensation [39]; and (iii) cells without DNA fragmentation and lower condensation (the latter two subpopulations both belonging to the brighter sperm population) (present study and References [39]). The origin of this nuclear heterogeneity is unclear, but we can speculate that it may reflect different stages/pathways of cell death. Chromatin condensation is a late step of germ cell apoptosis [41] which appears to be the most probable mechanism causing DNA fragmentation in non-viable spermatozoa [7]. Hence, it has been proposed that the low and high DNA condensed spermatozoa could represent, respectively, an early and a late step of the apoptotic program [39]. Conversely, brighter non-viable and non-DNA fragmented spermatozoa could be cells that died because of mechanisms different from apoptosis.

By multivariate analysis, we showed that the ability to predict fertile versus subfertile status is independent of semen quality, age, and abstinence time for all sDF parameters. This finding is expected for total [9,10] and viable sDF which does not associate to conventional semen parameters, abstinence time and age (present study) but not for non-viable sDF which correlates negatively with semen quality and positively with age and abstinence time (present study). This finding suggests that the amount of non-viable sDF reflects sperm/male traits affecting male fertility that are not revealed by routine semen analysis and are independent from age.

Given that viable and non-viable sDF similarly predict fertile versus subfertile status, one could expect that the predictive ability of total sDF was even higher. Intriguingly, this was not the case. This finding reinforces the hypothesis that the occurrence of DNA damage in viable and non-viable spermatozoa depends on a same sperm/male trait of vulnerability (for instance a failure of proper chromatin maturation), whatever is the mechanism inducing the damage.

The AUC values for the tested sDF parameters, obtained from ROC curves, were lower than those reported by previous similar studies [42,43], most probably because in our study the female factor of infertility was not excluded. Indeed, patients were recruited in a male population undergoing routine semen analysis, where female factors could be up to 40% [44]. Although recruiting patients instead of secure infertile males is a limitation of our study, it should be considered that such a limitation does not affect the comparison of the accuracy values between viable, non-viable, and total sDF which was the main aim of our study.

In conclusion, our study shows that the ability to predict fertile versus subfertile status of viable sDF is not different from that of total sDF as traditionally assessed or, surprisingly, of sDF in non-viable sperm population. The impact of viable DNA-fragmented spermatozoa appears to be related to their ability to reach and fertilize the oocyte. Conversely, non-viable sDF could represent a sign of a subtler damage extended beyond the non-viable cells, as suggested by the “tip of iceberg” theory, proposed by Evenson et al. [37].

## Figures and Tables

**Figure 1 jcm-09-01341-f001:**
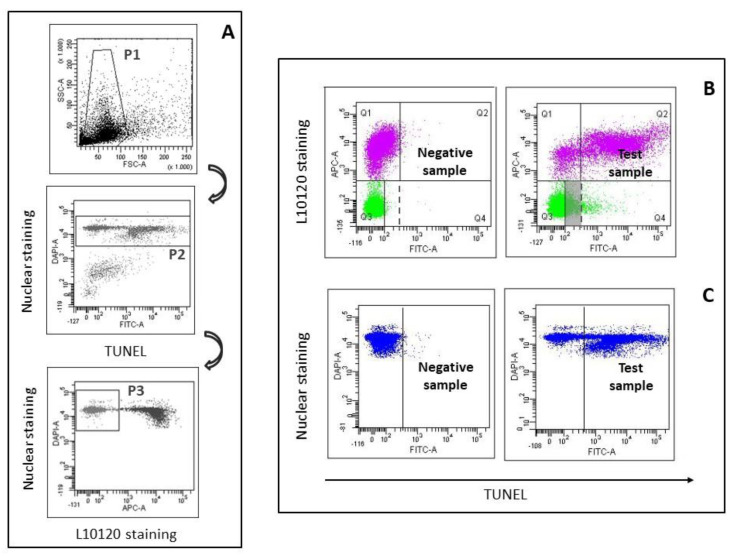
LiveTUNEL (terminal deoxynucleotidyl transferase (TdT)-mediated fluorescein-dUTP nick end labeling). (**A**) Gating strategy. Spermatozoa of native semen samples were gated within the Forward Scatter/Side Scatter flame shaped region (P1, upper panel) and then within the region containing DAPI-labelled events (P2, middle panel). Within spermatozoa, a further gate was drawn delimitating viable sperm (P3, lower panel) used during flow cytometric acquisition. (**B**) Typical L10120/TUNEL dot plots reporting DNA fragmentation in viable (green) and non-viable (pink) spermatozoa. Left panel: negative control, right panel: test sample. Note that non-specific fluorescence of the negative control (absence of TdT, left panel) was higher in non-viable than in viable spermatozoa. Grey region, fraction of viable sperm DNA fragmentation (sDF) that can be masked in traditional TUNEL assay. (**C**) Total sDF was calculated by gating total spermatozoa (blue), without discriminating viable and non-viable cells, in DAPI/TUNEL dot plots (left panel: negative control; right panel: test sample).

**Figure 2 jcm-09-01341-f002:**
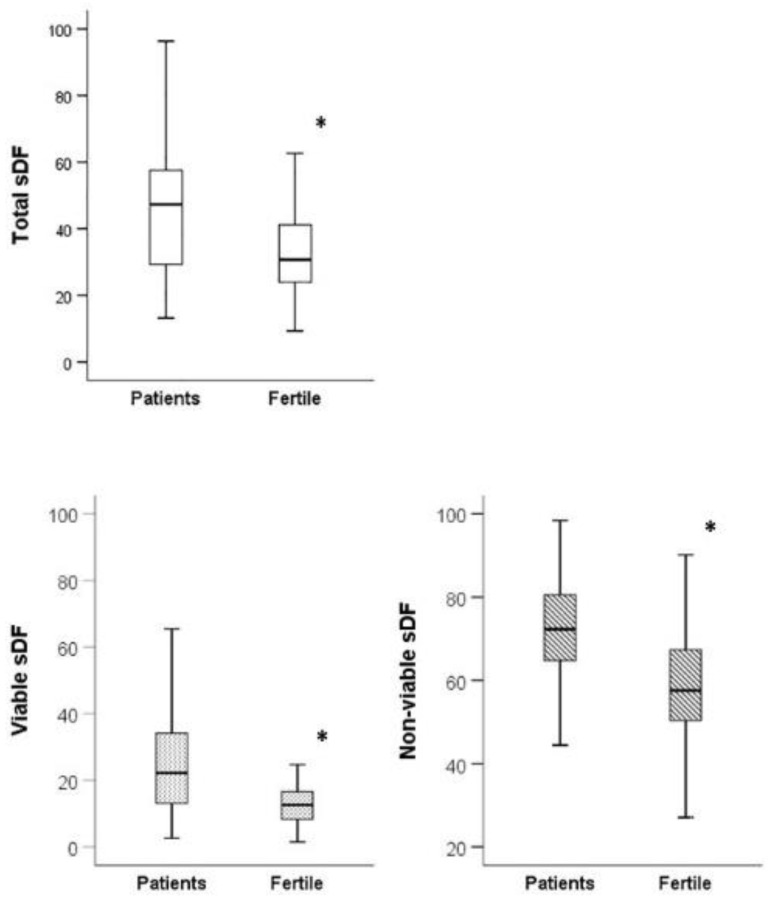
Total, viable, and non-viable sDF in patients and fertile men. Box graphs reporting median values (interquartile range) of the indicated sDF parameters in fertile men and in patients. * *p* < 0.001. Mann–Whitney U-test for independent data.

**Figure 3 jcm-09-01341-f003:**
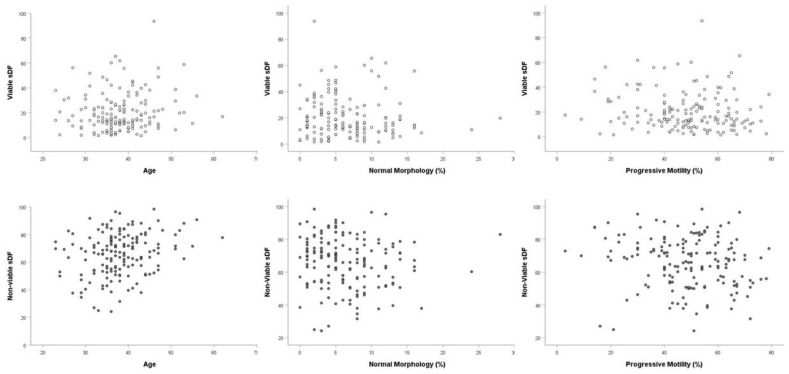
Relationship between sDF and semen quality and age. Dispersion plots reporting viable sDF (upper panels) and non-viable sDF (lower panels) against age, sperm morphology, and progressive motility as found in the 162 recruited subjects (91 patients and 71 fertile men).

**Table 1 jcm-09-01341-t001:** Age and semen parameters in fertile men and patients. Data are expressed as median (interquartile).

	All Subjects	Patients	Fertile Men	*p* ^#^
Number of Subjects	162	91	71	
Age (y)	38.00 (34.00–42.00)	38.00 (35.00–42.00)	36.00 (34.00–42.00)	0.121
Sperm Number (millions/ejaculate)	175.75 (106.35–271.80)	153.00 (67.84–266.60)	185.49 (114.40–285.00)	0.133
Concentration (10^6^/mL)	52.50 (33.00–102.20)	46.00 (22.00–88.00)	62.00 (38.60–121.50)	0.067
Total Motility (%)	60.00 (51.75–70.00)	57.00 (47.00–64.00)	65.00 (57.00–74.00)	<0.001
Progressive Motility (%)	50.00 (40.00-60.00)	45.00 (33.00–54.00)	54.00 (47.00–64.00)	<0.001
Morphology (%)	5.00 (3.00–9.00)	5.00 (2.00–8.00)	7.00 (4.00–9.00)	<0.001
Abstinence (d)	4.00 (3.00–4.25)	4.00 (3.00–5.00)	3.00 (3.00–4.00)	0.012
Semen Volume (ml)	3.10 (2.48–4.60)	3.20 (2.50–4.60)	3.10 (2.30–4.50)	0.125

^#^ Patients versus fertile men, Mann–Whitney U-test.

**Table 2 jcm-09-01341-t002:** Correlation analysis between sDF parameters and age, abstinence, and semen parameters in the recruited subjects (*n* = 162).

Variable	Total sDF	Viable sDF	Non−Viable sDF
	*r*	*r*	*r*
	*p*	*p*	*p*
Age, y	0.189 *	0.086	0.219 *
	0.016	0.278	0.005
Sperm Number (×10^6^/ejaculate)	−0.061	−0.088	0.022
	0.444	0.264	0.785
Concentration (10^6^/mL)	−0.03	−0.034	0.064
	0.705	0.67	0.418
Total Motility (%)	−0.277 *	−0.12	−0.202 *
	0	0.128	0.01
Progressive Motility (%)	−0.282 *	−0.136	−0.183 *
	0	0.084	0.02
Morphology (%)	−0.256 *	−0.101	−0.159 *
	0.001	0.202	0.044
Abstinence, d	0.203 *	0.074	0.231 *
	0.01	0.352	0.003

*r* = Spearman’s correlation coefficient; * statistically significant correlation.

**Table 3 jcm-09-01341-t003:** Area under the curve values of total, viable, and non-viable sDF for prediction model for fertile versus subfertile status.

	AUC (95% CI)	SE	*p*-Value
**Total sDF**	0.696 (0.615–0.776)	0.041	<0.001
**Viable sDF**	0.718 (0.640–0.797)	0.040	<0.001
**Non-viable sDF**	0.760 (0.685–0.835	0.038	<0.001

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
