# Peer review of "DNA Fragmentation in Viable and Non-Viable Spermatozoa Discriminates Fertile and Subfertile Subjects with Similar Accuracy"

_jcm, 2020, doi:10.3390/jcm9051341_

Round 1

Reviewer 1 Report

This is an interesting article by Muratori et al., examining the ability of sperm DNA fragmentation to predict male fertility. Specifically, the ability sperm DNA fragmentation in total, viable or nonviable spermatozoa to predict male fertility. Surprisingly, and contrary to what was expected, there was no difference in the predictive ability of these 3 populations, which is an important result, although it is not clear if this conclusion is supported by the data as the fertility status of the patients is not known.

 Comments

  1. By using male patients from couples where it is not clear if the infertility is due to the male or female partner, how do you know if the males are infertile. If the population includes fertile males how can you compare sDF between fertile and infertile males? Seems the conventional parameters in Table 1 could be used to exclude those males in the patient population who could be fertile and then determine whether this changes the conclusions.
  2. The title and conclusions indicate that DNA fragmentation can predict male fertility status. How can it predict male fertility status if the fertility status of the male patients in unknown?
  3. For Table 2 and figure 3, if sperm from fertile and patient groups are combined, how do you know there are no differences, especially for the viable sDF, that are missed? This should also be added.
  4. Lines 200-202, what relationship is being examined?
  5. I was not able to download supplemental table 1 and therefore, could not evaluate this data. This data needs to be reviewed.
  6. Discussion lines 253-255: this study does not determine where sperm DNA fragmentation occurs and thus cannot reinforce that it occurs outside the testis.

Reviewer 2 Report

The authors have addressed all concerns about statistical analysis, materials and method descriptions and clarity of discussion section.

The updated figure 1 is much improved. The discussion section has a greater contribution to the literature due to the clarity of the writing.

Inclusion of the abstinence time in the model and discussion strengthen the paper.

Thank you for your work.

Reviewer 3 Report

Comments on JCM-778302

The manuscript titled “DNA fragmentation in viable and non-viable spermatozoa predicts male fertility status with similar accuracy” by the authors Muratori et al., is an interesting study. Here the authors aimed to identify novel aspects of sperm DNA fragmentation in live/dead sperm to determine the fertility status of men. This is a novel/unique study where two staining methods were used to simultaneously decipher the presence or absence of sperm DNA fragmentation in the sperm population. This study has both merits and drawbacks and suggestions to improve, that are discussed below.

Merits:

  1. The technology used here is novel and unique, where double staining method for TUNEL assay is used. The use of flowcytometer to analyze the results is a merit.
  2. The sample size is large enough to provide sufficient power for statistical analysis.
  3. The aim of the study, hypothesis seams rational and has been addressed perfectly in this manuscript.

Disadvantages

  1. The patient group used in this study (n = 91), were male partners of infertile couple and not all of these cases may have actual male infertility issues. Therefore, there is a bias comparing the patient group with the fertile donor group.
  2. Throughout the manuscript, the authors claim that the viable sperm population is the clinically important sperm population and hence DNA fragmentation in viable sperm is critical. However, viable and progressively motile sperm population is practically the clinical important sperm subset, which is able to reach and fertilize the oocyte. The authors could have used a small portion of semen to perform density gradient or swim-up method to isolate the progressively motile sperm, as the semen volume averages 3.2 ml.
  3. This study has one aim (which is fully addressed) and lacks any clinical data, either IUI or IVF outcomes to substantiate the effects of sperm DNA fragmentation is a drawback.

Suggestions

I have couple of suggestions that could be included to enhance the clinical application of this study.

  1. Based on the semen analysis data presented in this study, it is possible to determine the patient population into fertile and infertile group using the World Health Organization (2010) criteria of semen profiling, which is the current gold standards followed in all andrology clinics. Once the groups were determined, it’s possible to perform the comparison of DNA fragmentation in viable vs. nonviable sperm between the normal and abnormal semen profile groups, and discuss the clinical implications? Also, clinically useful in cases with idiopathic male infertility. 
  2. Most clinics worldwide, including my laboratory uses ~20% as a cutoff value for TUNEL assay and patient population above the cutoff value are considered abnormal and have a reduced ability for ART success. Is it possible to identify normal and abnormal sperm DNA fragmentation groups (based on the cutoff value) within the patient/fertile population and compare the level of DNA fragmentation is viable vs. nonviable sperm?

Round 2

Reviewer 1 Report

This is a nice and interesting article by Muratori et al., examining sperm DNA fragmentation in total, viable or nonviable spermatozoa. Surprisingly, and contrary to what was expected, there was no difference in the predictive ability of these 3 populations, which is an important result. The authors have done a nice job addressing the reviewer’s comments. Thank you.